

# AlphaDDA: strategies for adjusting the playing strength of a fully trained AlphaZero system to a suitable human training partner

Kazuhisa Fujita[1,2]

[1] Komatsu University, Komatsu, Ishikawa, Japan
[2] University of Electro-Communications, Chofu, Tokyo, Japan

## ABSTRACT

Artificial intelligence (AI) has achieved superhuman performance in board games such as Go, chess, and Othello (Reversi). In other words, the AI system surpasses the level of a strong human expert player in such games. In this context, it is difficult for a human player to enjoy playing the games with the AI. To keep human players entertained and immersed in a game, the AI is required to dynamically balance its skill with that of the human player. To address this issue, we propose AlphaDDA, an AlphaZero-based AI with dynamic difficulty adjustment (DDA). AlphaDDA consists of a deep neural network (DNN) and a Monte Carlo tree search, as in AlphaZero. AlphaDDA learns and plays a game the same way as AlphaZero, but can change its skills. AlphaDDA estimates the value of the game state from only the board state using the DNN. AlphaDDA changes a parameter dominantly controlling its skills according to the estimated value. Consequently, AlphaDDA adjusts its skills according to a game state. AlphaDDA can adjust its skill using only the state of a game without any prior knowledge regarding an opponent. In this study, AlphaDDA plays Connect4, Othello, and 6x6 Othello with other AI agents. Other AI agents are AlphaZero, Monte Carlo tree search, the minimax algorithm, and a random player. This study shows that AlphaDDA can balance its skill with that of the other AI agents, except for a random player. AlphaDDA can weaken itself according to the estimated value. However, AlphaDDA beats the random player because AlphaDDA is stronger than a random player even if AlphaDDA weakens itself to the limit. The DDA ability of AlphaDDA is based on an accurate estimation of the value from the state of a game. We believe that the AlphaDDA approach for DDA can be used for any game AI system if the DNN can accurately estimate the value of the game state and we know a parameter controlling the skills of the AI system.

# INTRODUCTION

Artificial intelligence (AI) has witnessed rapid advances in recent years and has shown remarkable applicability in various fields. One of the research domains in AI is developing a superhuman-level AI for playing board games. Researchers have demonstrated that AI can defeat top-ranked human players in certain board games such as checkers, chess (*Campbell, Hoane & Hsu, 2002*), and Othello (*Buro, 1997*). In 2016, AlphaGo (*Silver et al.,*

Corresponding author
Kazuhisa Fujita,
kazu@spikingneuron.net

*2016*), which is Go-playing AI, defeated one of the top Go players; however, the best Go software did not even have a decent chance against average amateur players before 2016 (*Li & Du, 2018*). Furthermore, AlphaZero (*Silver et al., 2018*) can play various board games and perform better than other superhuman-level AIs at chess, Shogi, and Go. Thus, in traditional board games, game-playing AI has already been shown to demonstrate superhuman skills.

However, the problem is that a game-playing AI with superhuman skills is too strong for general human players. When a human plays against an AI that is too strong, it easily defeats the human player, leading to frustration. In contrast, an AI that is too weak is also not desirable because a human player can easily defeat the AI, and thus the player will not feel challenged. In both cases, a human player will consequently leave the game.

In order for the AI system to become a suitable training partner and continue attracting them to a game, the AI is required to balance its playing skills with that of a human player. According to *Segundo, Calixto & de Gusmão (2016)*, this balancing is critical in ensuring a pleasant gaming experience for the human player. The real-time balancing is called dynamic difficulty adjustment (DDA). DDA is a method of automatically modifying a game's features, behaviors, and scenarios in real time, depending on the human player's skill so that the human player does not feel bored or frustrated (*Zohaib & Nakanishi, 2018*). In particular, it is important to adjust an AI's skill when a human plays a game with the AI. DDA requires that an AI or a game system evaluate the human player's skill and/or the game's state and adapt its skill to the opponent's skill as quickly as possible (*Andrade et al., 2006*). However, it is difficult to estimate a player's skill in real time because there is great diversity among players in terms of skill and strategies adopted in a game. In addition, the evaluation of the game state is difficult for board games. In a role-playing or fighting game, a player or game system can capture the values directly, representing whether the state is good or bad. For example, a hit point in these games is the maximum amount of damage caused and represents a character's health. However, in board games, players can obtain only the board state as information about the game during the game, and an AI is required to evaluate the game state from only the board state. Furthermore, the game state and its value depend on the player's skill. Thus, an AI estimates a player's skill or evaluates the game state from only the board state and must adjust its skill to that of the opponent using the estimation and evaluation in board games.

In this study, we propose AlphaDDA, which is a game AI that adapts its skill dynamically according to the game state. AlphaDDA is based on AlphaZero and consists of a Monte Carlo tree search (MCTS) and a deep neural network (DNN). AlphaDDA estimates the value of the game state from the board state using a DNN. It then adjusts its skill to that of the opponent by changing its skill according to the estimated value. This study presents the AlphaDDA algorithm, and demonstrates its ability to adjust its skill to that of an opponent in board games.

## RELATED WORK

AI game players have obtained superhuman-level skills for traditional board games such as checkers (*Schaeffer et al., 1993*), Othello (*Buro, 1997, 2003*), and chess (*Campbell, 1999*;

*Hsu, 1999*; *Campbell, Hoane & Hsu, 2002*). In 2016, the Go-playing AI, AlphaGo (*Silver et al., 2016*), defeated the world's top player in Go and became the first superhuman-level Go-playing algorithm. AlphaGo relies on supervised learning from a large database of expert human moves and self-play data. AlphaGo Zero (*Silver et al., 2017*), in turn, defeated AlphaGo, without requiring to prepare a large training dataset. In 2018, *Silver et al. (2018)* proposed AlphaZero, which does not have restrictions on playable games. AlphaZero has outperformed other superhuman-level AIs at Go, Shogi, and chess. However, AlphaZero is not suitable as an opponent of human players because AlphaZero is too strong for almost all human players.

Some researchers have shown that AlphaZero can learn games other than two-player perfect information games. *Hsueh et al. (2018)* have investigated that AlphaZero can play a non-deterministic game. *Moerland et al. (2018)* have extended AlphaZero to deal with continuous action space. *Petosa & Balch (2019)* have modified AlphaZero to support a multiplayer game. *Schrittwieser et al. (2020)* have developed MuZero. MuZero (*Schrittwieser et al., 2020*) is an AlphaZero-based tree search with a learned model that achieves superhuman performance in board games and computer games without any knowledge of their game rules.

Some researchers have improved AlphaZero algorithm. *Wang, Preuss & Plaat (2020, 2021)* have used MCTS with warm-start enhancements in self-play and improved the quality of game playing records generated by self-play. They have achieved improving Elo-rating of AlphaZero.

A player will not feel challenged if the game is either too easy or too difficult for them. In the absence of a sense of being challenged, it is expected that the players will quit the game after playing it a few times. For a game to attract human players, the game's difficulty or the skill of a game-playing AI agent must be balanced with the human player's strength. The enjoyment of human players depends on the balance between the game's difficulty and their skill. The game development community recognizes game balancing as a key characteristic of a successful game (*Rouse, 2000*). One approach for difficulty adjustment is that the player selects a few static difficulty levels (for example, easy, medium, and hard) before the game starts (*Anagnostou & Maragoudakis, 2012*; *Andrade et al., 2006*; *Glavin & Madden, 2018*; *Sutoyo et al., 2015*). The selected difficulty has a direct and usually static effect on the skill of non-player characters (AIs). However, the difficulty is selected with no real comprehension of how well the difficulty level suits the player's abilities. This method for difficulty adjustment produces another problem, namely, static difficulties. The skill of AIs is too static to flexibly match the competence of every opponent. Player abilities are not limited to a few levels and increase with the experience of gameplay. DDA has been proposed to solve these problems. DDA can help game designers provide more attractive, playable, and challenging game experiences for players (*Sutoyo et al., 2015*).

DDA is the process of dynamically adjusting the level of difficulty involving an AI's skill according to the player's ability in real time in a computer game. The goal of DDA is to ensure that the game remains challenging and can cater to many different players with varying abilities (*Glavin & Madden, 2018*). Many researchers have already proposed many DDA methods. *Zook & Riedl (2012)* focused on the challenge-tailoring problem in

adventure role-playing games. They modeled a player and developed an algorithm to adapt the content based on that model. *Sutoyo et al. (2015)* used players' lives, enemies' health, and skill points to determine the multipliers that affect the game's difficulties. The multipliers are changed at the end of every level where the change points depend on the performance gameplay of the players. For instance, if the players use a good strategy and do not lose any lives at certain levels, the multiplier points will be increased, and the next levels will be more challenging. *Lora et al. (2016)* estimated a player's skill level by clustering the cases describing how the player places a sequence of consecutive pieces in the game board in Tetris. *Xue et al. (2017)* modeled players' in-game progress as a probabilistic graph consisting of various player states. The transition probabilities between states depend on the difficulties in these states and are optimized to maximize a player's engagement. *Pratama & Krisnadhi (2018)* developed a DDA method for a turn-based role-playing game. They modeled the enemy such that a player could have an approximately equal chance of winning and losing against the enemy. The enemy seeks the action that leads it to win or lose in the game with as little margin as possible using MCTS. *Ishihara et al. (2018)* proposed the Upper Confidence Bound (UCB) score to balance the skills of a human player and an AI in a fighting game. Their proposed UCB score depends on the hit points of the player and its opponent. The AI selects an action using MCTS based on the proposed UCB score.

## ALPHADDA

This study proposes AlphaDDA, which is a game-playing AI with DDA. The algorithm of AlphaDDA is straightforward. AlphaDDA estimates a value indicating the winner from the board state of a game and changes its skill according to this value.

AlphaDDA is based on AlphaZero proposed by *Silver et al. (2018)*. AlphaZero uses a combination of a DNN and an MCTS (Fig. 1A). The DNN estimates the value and the move probability from the board state. MCTS searches a game tree using the value and the move probability. There are three reasons for applying AlphaZero. First, AlphaZero has no restrictions on the board games that it can play. Second, AlphaZero has sufficient playing skills to amuse high-level human players. Third, AlphaZero can precisely evaluate the current state of a game as a value. The details of AlphaZero are provided in "AlphaZero".

AlphaDDA consists of a DNN and MCTS, similar to AlphaZero. The DNN has a body and two heads: the value head and the policy head. The body is based on ResNet and has residual blocks. The value head and the policy head output the value and policy (the probability of selecting a move), respectively. AlphaDDA decides a move based on MCTS with upper confidence bound for tree search (UCT) using the value and policy. AlphaDDA uses the same DNN and the same MCTS algorithm as AlphaZero. The details of the DNN and MCTS algorithm are described in "AlphaZero".

The algorithm and learning method of AlphaDDA are the same as those of AlphaZero. In other words, AlphaDDA is an addition of DDA to the trained AlphaZero. Thus, we trained AlphaZero and directly used the weights of the trained DNN of AlphaZero as those of AlphaDDA's DNN. The parameters of AlphaDDA are the same as those of AlphaZero and are listed in "AlphaZero".

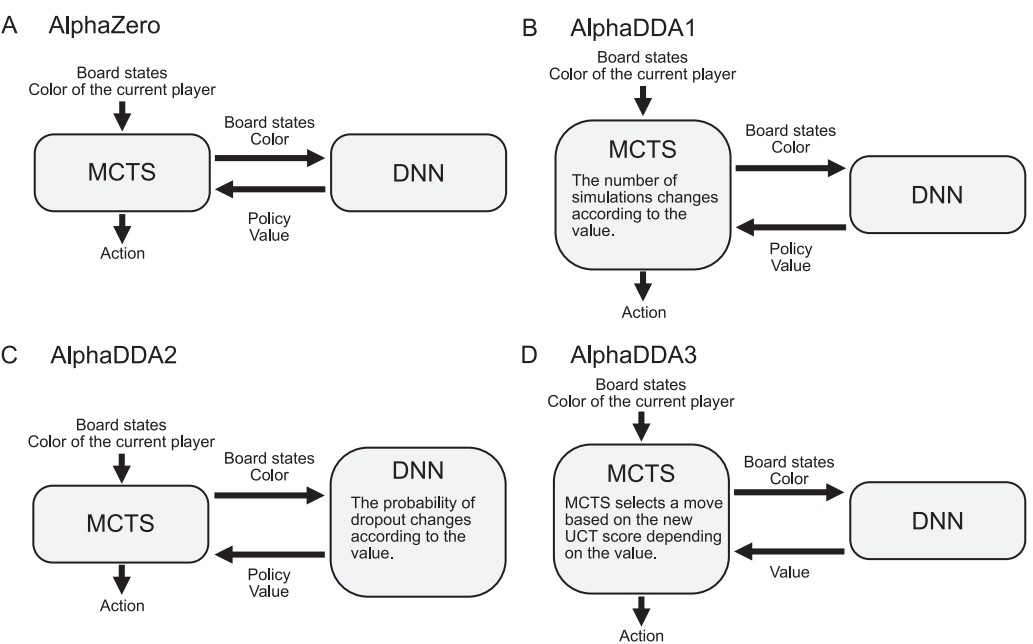

**Figure 1 Flows for AlphaZero and AlphaDDAs.** (A) Flow for vanilla AlphaZero. (B) Flow for AlphaDDA1. AlphaDDA1 changes its skill by varying the number of simulations according to the value. (C) Flow for AlphaDDA2. The DNN is damaged by dropout because its probability changes according to the value. (D) Flow for AlphaDDA3. The MCTS with the new score prefers a worse move when the DNN predicts that AlphaDDA3 will win in the current state and *vice versa*.

AlphaDDA does not always select the best move, even though it is based on AlphaZero. AlphaDDA selects a worse move, which is not the best move, when the DNN predicts that AlphaDDA wins and *vice versa*. Let us consider that the opponent is a lower-skill player. The opponent selects a worse move, and then the DNN predicts the win of AlphaDDA from the state of the game. In this case, if AlphaDDA continues to select the best move, it will beat the opponent with a high probability. In order to reduce AlphaDDA's probability of winning, AlphaDDA selects a worse move. AlphaDDA's selection of the worse move will increase the opponent's probability of winning.

Three AlphaDDAs are proposed herein: AlphaDDA1, AlphaDDA2, and AlphaDDA3. AlphaDDA1 changes the number of simulations according to the value (Fig. 1B). AlphaDDA1 can adjust its skill to that of an opponent by changing the number of simulations. AlphaDDA2 changes the probability of dropout according to the value (Fig. 1C). The dropout refers to stochastically ignoring some units in the DNN, which damages the DNN. AlphaDDA2 selects a worse move because the damaged DNN provides an inaccurate output. Thus, AlphaDDA2 can adjust its strength according to that of an opponent using the dropout. AlphaDDA3 adopts the new UCT score depending on the value (Fig. 1D). The MCTS with the new score prefers a worse move when the DNN predicts that AlphaDDA3 wins from the current state and *vice versa*.

## The value of the board state

AlphaDDA obtains the value of the board state from the DNN. The DNN estimates the value and the move probability from the board state. The input of the DNN comprises the $T$ board states immediately before placing the disc at the $n$th turn. The value at the $n$th turn $v_n$ is a continuous variable ranging from 1 to $-1$. When $v_n$ is closer to the disc color of AlphaDDA $c_{\text{AlphaDDA}}$, the DNN assumes that the winning probability of AlphaDDA is higher. $c_{\text{AlphaDDA}}$ is 1 and $-1$ when AlphaDDA is the first player and the second player, respectively. In this study, AlphaDDA uses the mean of the value $\bar{v}_n$ defined as

$$\bar{v}_n = \frac{1}{N_h} \sum_{i=0}^{N_h-1} (v_{n-i}), \tag{1}$$

where $N_h$ is the number of the values. If $n - N_h < 0$, $N_h = n$. AlphaDDA adjusts its skill to that of an opponent by changing its skill according to $\bar{v}_n$.

## AlphaDDA1

AlphaDDA1 balances its skill and that of the opponent by changing the number of simulations $N_{\text{sim}}(\bar{v}_n)$ according to the value $\bar{v}_n$. AlphaDDA1 is AlphaZero with a variable number of simulations depending on $\bar{v}_n$. The quality of a move selected by the MCTS depends on $N_{\text{sim}}(\bar{v}_n)$. Thus, AlphaDDA1 can change its skill according to $\bar{v}_n$. The number of simulations $N_{\text{sim}}(\bar{v}_n)$ is defined as

$$N_{\text{sim}}(\bar{v}_n) = \left\lceil 10^{-A_{\text{sim}}(\bar{v}_n c_{\text{AlphaDDA}} + B_{\text{sim0}})} \right\rceil, \tag{2}$$

$$N_{\text{sim}}(\bar{v}_n) \leftarrow \begin{cases} N_{\max} & \text{if } N_{\text{sim}}(\bar{v}_n) > N_{\max} \\ N_{\text{sim}}(\bar{v}_n) & \text{otherwise} \end{cases}, \tag{3}$$

where $A_{\text{sim}}$, $B_{\text{sim0}}$, and $N_{\max}$ are hyperparameters. $\lceil \cdot \rceil$ is a ceiling function. $N_{\text{sim}}(\bar{v}_n)$ is always greater than or equal to 1 because $\lceil 10^x \rceil \geq 1$, where $x \in R$. The base is not limited to 10 because if $N_{\text{sim}}(\bar{v}_n)$ increases exponentially, that is sufficient.

$N_h$, $A_{\text{sim}}$, $B_{\text{sim0}}$, and $N_{\max}$ are set as the values shown in Table 1. These values are determined using a grid search such that the difference in the win and loss rates of AlphaDDA1 against the other AIs is minimal. The details of the grid search are shown in "Grid search".

## AlphaDDA2

AlphaDDA2 changes its skill by damaging the DNN using dropout. Dropout refers to ignoring some units selected at random and is generally used while training a DNN to avoid overfitting. In AlphaDDA2, dropout is used to damage the DNN. In other words, AlphaDDA2 is AlphaZero with the DNN damaged by dropout. The damaged DNN provides a more inaccurate output. Consequently, AlphaDDA2 becomes weaker owing to the inaccurate output. A larger probability of dropout causes more inaccurate outputs and makes AlphaDDA2 weaker.

**Table 1 Parameters of AlphaDDA1.**

| Game | $N_h$ | $A_{sim}$ | $B_{sim0}$ | $N_{max}$ |
|---|---|---|---|---|
| Connect4 | 3 | 2.0 | −1.4 | 200 |
| 6x6 Othello | 4 | 1.4 | −1.6 | 200 |
| Othello | 2 | 2.8 | −1.4 | 400 |

AlphaDDA2 changes the probability of dropout $P_{drop}$ according to the value calculated by the DNN with $P_{drop} = 0$. The value is calculated by the DNN not damaged by the dropout. The dropout is inserted into the fully connected layers in the value and policy heads, as shown in Fig. 2. $P_{drop}$ changes with the mean value $\bar{v}_n$ as follows:

$$P_{drop}(\bar{v}) = A_{drop}(\bar{v}_n + P_{drop0}), \tag{4}$$

$$P_{drop}(v) \leftarrow \begin{cases} 0 & \text{if } P_{drop}(\bar{v}_n) < 0 \\ P_{max} & \text{if } P_{drop}(\bar{v}_n) > P_{max} , \\ P_{drop}(\bar{v}_n) & \text{otherwise} \end{cases} \tag{5}$$

where $A_{drop}$ and $P_{drop0}$ are hyperparameters.

$N_h$, $A_{drop}$, and $P_{drop0}$ are set as the values shown in Table 2. These values are determined using a grid search such that the difference in the win and loss rates of AlphaDDA2 against the other AIs is minimal. The details of the grid search are shown in "Grid search". $P_{max}$ is set as 0.95. This limit of $P_{drop}(v)$ is required to avoid the outputs of all the units in the DNN being 0.

## AlphaDDA3

AlphaDDA3 adjusts its skill to that of an opponent using the new UCT score based on two ideas. The first is that AlphaDDA3 assumes that an opponent generates a board state with a mean value of $\bar{v}_n$ because the value depends on the player's skill. For example, if an opponent is weaker, it continues to select a worse move, and the state remains bad for it. Thus, the DNN continues to predict that AlphaDDA3 beats the opponent. Second, AlphaDDA3 selects a better or a worse move if the value is further from or closer to $c_{AlphaDDA}$, respectively. A value further from $c_{AlphaDDA}$ means that the DNN predicts that the opponent beats AlphaDDA3 and *vice versa*. The UCT score based on these two ideas is denoted by

$$U(s_t, a) = W(s_t, a)/N(s_t) + C\sqrt{2\ln(N(s_t) + 1)/(n(s_t, a) + 1)}, \tag{6}$$

$$W(s_t, a) \leftarrow W(s_t, a) - |v(s_t) + \bar{v}_n c(s_t, a)c_{AlphaDDA}|, \tag{7}$$

where $W(s_t, a)$ is the cumulative value, $N(s_t)$ is the visit count of the node $s_t$, $n(s_t, a)$ is the visit count of the edge $(s_t, a)$, $C$ is the exploration rate, $v(s_t)$ is the value of state $s_t$, $c_{AlphaDDA}$ is the disc color of AlphaDDA3, and $c(s_t, a)$ is the disc color of the player for $(s_t, a)$. The absolute value of the difference between $v(s_t)$ and $\bar{v}_n$ is subtracted from $W(s_t, a)$ if the player for $(s_t, a)$ is the opponent. This subtraction means that the opponent selects a move such that the board state provides the same value as the mean value $\bar{v}_n$. The absolute

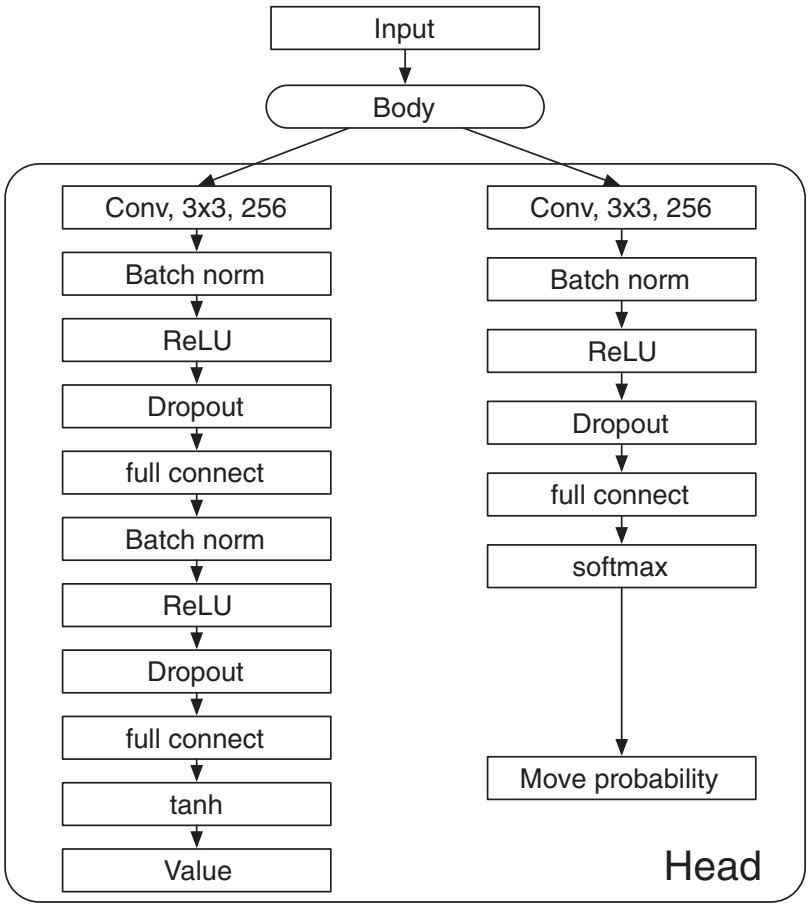

**Figure 2 Architecture of the deep neural network in AlphaDDA2.**

value of the sum of $v(s_t)$ and $\bar{v}_n$ is subtracted from $W(s_t, a)$ if the player for $(s_t, a)$ is AlphaDDA3. The subtraction means that AlphaDDA3 prefers the move in which the sum of $v(s_t)$ and $\bar{v}_n$ is 0. In other words, AlphaDDA3 prefers the move in which $v(s_t)$ is $-\bar{v}_n$. This preference means that AlphaDDA3 selects a worse move if $\bar{v}_n$ is closer to $c_{\text{AlphaDDA}}$ and *vice versa*.

The hyperparameters $N_h$ and $C$ are set as the values shown in Table 3. The values of $N_h$ and $C$ are determined using a grid search such that the difference in the win and loss rates of AlphaDDA3 against the other AIs is minimal. The details of the grid search are shown in "Grid search".

## EXPERIMENTAL SETTINGS

### Games
In this study, AlphaDDAs play Connect4, 6x6 Othello, and Othello. These games are two-player, deterministic, and zero-sum games of perfect information. Connect4 was published by Milton and is a two-player connection game. The players drop discs into a $7 \times 6$ board. When a player forms a horizontal, vertical, or diagonal line of its four discs, the player wins. Othello (Reversi) is also a two-player strategy game that uses an $8 \times 8$

**Table 2 Parameters of AlphaDDA2.**

| Game | $N_h$ | $A_{drop}$ | $P_{drop0}$ |
| --- | --- | --- | --- |
| Connect4 | 1 | 10 | −0.2 |
| 6x6 Othello | 4 | 5 | −0.5 |
| Othello | 4 | 20 | −0.9 |

**Table 3 Parameters of AlphaDDA3.**

| Game | $N_h$ | $C$ |
| --- | --- | --- |
| Connect4 | 5 | 1.25 |
| 6x6 Othello | 1 | 0.75 |
| Othello | 5 | 0.6 |

board. In Othello, the disc is white on one side and black on the other. Players take turns placing a disc on the board with their assigned color facing up. During a play, any discs of the opponent's color are turned over to the current player's color if they are in a straight line and bounded by the disc just placed and another disc of the current player's color. 6x6 Othello is Othello using a 6x6 board.

## Opponents

In this study, eight AIs, AlphaZero, MCTS1, MCTS2, MCTS3, MCTS4, Minimax1, Minimax2, and Random, were used to evaluate the performance of AlphaDDA. AlphaZero was trained as described in "Alphazero". MCTS1, MCTS2, MCTS3, and MCTS4 are MCTS methods with 300, 100, 200, and 50 simulations, respectively. In all the MCTSs, the child nodes of the node at five visits are opened. Minimax1 and Minimax2 are Minimax methods that select a move using the minimax algorithm based on an evaluation table. The details of MCTS and Minimax are described in "Monte carlo tree search" and "Minimax algorithm", respectively. Random selects a move from valid moves uniformly at random. AlphaZero, MCTS1, MCTS2, Minimax1, and Random are used for the grid search ("Grid search"), calculation of Elo rating ("Elo rating of board game playing algorithms"), and investigation of the dependence of the strength of AlphaZero on the parameters ("Dependence of the strength of AlphaZero on its parameters" and "Dependence of Elo rating on the probability of dropout"). All the AIs are used to evaluate the performances of AlphaDDAs. In particular, MCTS3, MCTS4, and Minimax2 are only used to evaluate the performance of AlphaDDAs and not used for the grid search.

## Software

The Python programming language and its libraries, NumPy for linear algebra computation and PyTorch for deep learning, were used. The program codes of AlphaDDA, AlphaZero, MCTS, Minimax, and the board games are available on GitHub (https:// github.com/KazuhisaFujita/AlphaDDA).

**Table 4 Elo rating.**

| Game | Connect4 | 6x6 Othello | Othello |
|---|---|---|---|
| AlphaZero | 1,978 | 1,960 | 1,970 |
| MCTS1 | 1,731 | 1,681 | 1,618 |
| MCTS2 | 1,506 | 1,455 | 1,534 |
| Minimax1 | 1,355 | 1,425 | 1,422 |
| Random | 930.6 | 979.5 | 953.6 |

## RESULTS

### Elo rating of board game playing algorithms

This subsection shows the relative strengths of five AI agents: AlphaZero, MCTS1, MCTS2, Minimax1, and Random. The Elo rating is used to evaluate the relative strength of each AI. The Elo rating is calculated through 50 round-robin tournaments between the agents. The Elo ratings of all the agents are initialized to 1,500. The details of the Elo rating calculation are provided in "Elo rating".

Table 4 shows the Elo ratings of the agents for Connect4, 6x6 Othello, and Othello. AlphaZero is the strongest player. This result shows that AlphaZero masters all the games well and is stronger than the other agents. MCTS1 shows the second-best performance. Random is the worst player for all the games.

### Dependence of the strength of AlphaZero on its parameters

The skill of AlphaZero depends on the values of its parameters. To effectively change AlphaZero's skill by varying its parameters, the dependence of the skill on each parameter needs to be investigated. This subsection examines the relationship between the Elo rating of AlphaZero and the values of two parameters: the number of simulations $N_{sim}$ and $C_{puct}$. $N_{sim}$ is the number of times a game tree is searched in MCTS and directly affects the decision of a move. $C_{puct}$ is a parameter in the UCT score and affects the tree search. The DNN of AlphaZero with the changed parameter has the same weights as those of AlphaZero used in "Elo rating of board game playing algorithms". In other words, AlphaZero with the changed parameter is already trained. The opponents are AlphaZero, MCTS1, MCTS2, Minimax1, and Random. The opponents' Elo ratings were fixed to the values listed in Table 4. AlphaZero with the changed parameter plays 50 games with each opponent (25 games when AlphaZero with the changed parameter is the first player and 25 games when AlphaZero with the changed parameter is the second player).

Figure 3A shows that the Elo rating depends on $N_{sim}$ for all the games. The Elo ratings increase with $N_{sim}$ for all the games. For Connect4, the Elo rating quickly increases from 5 to 20 simulations. No significant change in the Elo rating is observed from 40 simulations for Connect4. For 6x6 Othello and Othello, the Elo ratings increase with $N_{sim}$ and do not change significantly from 200 simulations. These results show that the skill of AlphaZero depends on $N_{sim}$. However, AlphaZero is not weak enough even though $N_{sim} = 1$ because the Elo rating of AlphaZero is still larger than that of Random when $N_{sim} = 1$.

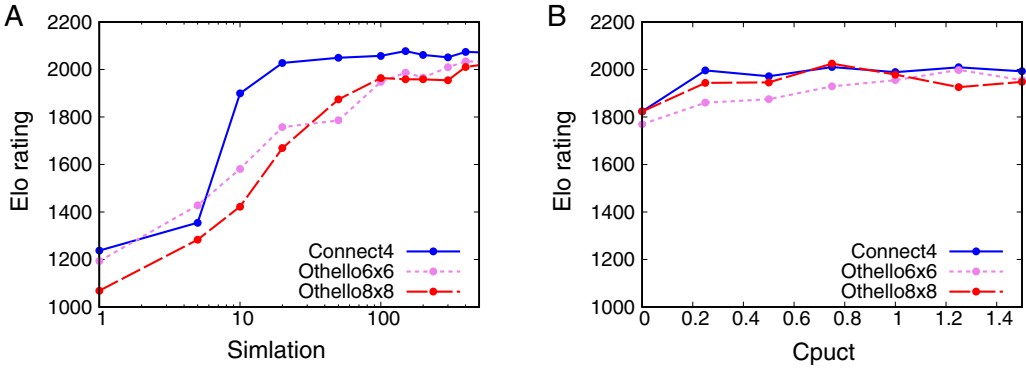

**Figure 3 Relationship between Elo ratings and parameters.** (A) Dependence of the Elo rating of AlphaZero on the number of simulations $N_{\text{sim}}$. (B) Dependence of the Elo rating of AlphaZero on $C_{\text{puct}}$.

Figure 3B shows the dependence of the Elo rating on $C_{\text{puct}}$. The Elo rating does not change significantly with $C_{\text{puct}}$ apart from $C_{\text{puct}} = 0$ for all the games. This result suggests that we cannot control the skill of AlphaZero by changing $C_{\text{puct}}$.

## Dependence of Elo rating on the probability of dropout

This subsection examines the change in the skill of AlphaZero with the DNN damaged by dropout. Dropout refers to ignoring units selected at random and is used to render the value and policy inaccurate in this experiment. A dropout was inserted into the head part of the DNN, as shown in Fig. 2. The DNN weights of AlphaZero with dropout are the same as those of AlphaZero used in "Elo rating of board game playing algorithms". AlphaZero with dropout plays games with AlphaZero, MCTS1, MCTS2, Minimax, and Random. The opponent's Elo ratings were fixed to the values in Table 4. AlphaZero with dropout plays 50 games with each opponent (25 games when AlphaZero with dropout is the first player and 25 games when AlphaZero with dropout is the second player).

Figure 4 shows the dependence of the Elo rating of AlphaZero with dropout on the probability of dropout. The Elo ratings of AlphaZero with dropout decrease with the probability of dropout. This result shows that we can weaken AlphaZero using dropout. However, AlphaZero with dropout is still stronger than Random, even though the probability of dropout is 0.95.

## Evaluation of AlphaDDAs

This subsection tests the ability of AlphaDDAs to adapt to an individual opponent in Connect4, 6x6 Othello, and Othello. AlphaDDAs compete with the AI agents: AlphaZero, MCTS1, MCTS2, MCTS3, MCTS4, Minimax1, Minimax2, and Random. The weights of AlphaDDA and AlphaZero are the same as those of AlphaZero used in "Elo rating of board game playing algorithms". The win-loss-draw rate is used for an metric to evaluate AlphaDDA's DDA performances. The win-loss-draw rate is a good metric for deciding whether AlphaDDA can adjust its skill to an opponent. AI's skill that is appropriate for an

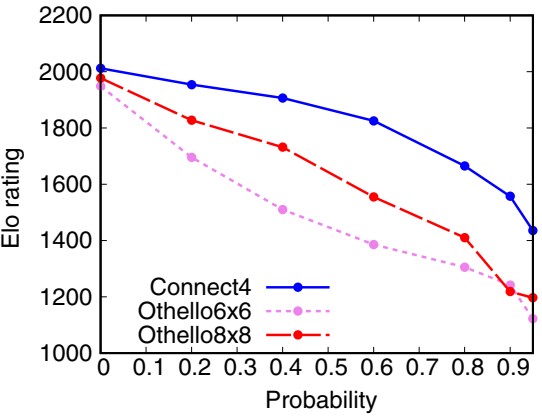

**Figure 4** Dependence of the Elo rating of AlphaZero on the probability of dropout.

**Table 5** Win, loss, and draw rates of AlphaDDAs for Connect4.

| Opponent | AlphaDDA1 | | | AlphaDDA2 | | | AlphaDDA3 | | |
|---|---|---|---|---|---|---|---|---|---|
| | Win | Loss | Draw | Win | Loss | Draw | Win | Loss | Draw |
| AlphaZero | 0.02 | 0.02 | 0.96 | 0.01 | 0.19 | 0.8 | 0.0 | 0.72 | 0.28 |
| MCTS1 | 0.22 | 0.76 | 0.02 | 0.29 | 0.67 | 0.04 | 0.0 | 1.0 | 0.0 |
| MCTS2 | 0.46 | 0.54 | 0.0 | 0.55 | 0.42 | 0.03 | 0.01 | 0.98 | 0.01 |
| MCTS3 | 0.32 | 0.63 | 0.05 | 0.39 | 0.52 | 0.09 | 0.0 | 0.98 | 0.02 |
| MCTS4 | 0.58 | 0.41 | 0.01 | 0.79 | 0.2 | 0.01 | 0.02 | 0.95 | 0.03 |
| Minimax1 | 0.69 | 0.3 | 0.01 | 0.73 | 0.27 | 0.0 | 0.0 | 0.99 | 0.01 |
| Minimax2 | 0.68 | 0.32 | 0.0 | 0.94 | 0.06 | 0.0 | 0.04 | 0.96 | 0.0 |
| Random | 0.98 | 0.02 | 0.0 | 1.0 | 0.0 | 0.0 | 0.46 | 0.49 | 0.05 |

opponent is one in which the player has the same chance to either win or lose the game (*Demediuk et al., 2017*) or the draw rate is 1.

## Connect4

Table 5 shows the win, loss, and draw rates of AlphaDDAs against the AI agents for Connect4. The win-loss-draw rate is calculated through 100 games with each AI (50 games when AlphaDDA is the first player and 50 games when AlphaDDA is the second player).

AlphaDDA1 is on par with AlphaZero, MCTS2, and MCTS4. AlphaDDA1 is weaker than MCTS1 and MCTS3 but can adjust its skill against the opponents. AlphaDDA1 is stronger than Minimax1 and Minimax2 but can adjust its skill against the opponents. Thus, AlphaDDA1 achieves DDA against AlphaZero, MCTS1, MCTS2, MCTS3, MCTS4, Minimax1, and Minimax2. AlphaDDA1 can adjust its strength against the AI agents not used for grid search. However, AlphaDDA1 has an extremely high win rate against Random.

AlphaDDA2 is on par with AlphaZero and MCTS2. AlphaDDA2 is weaker than MCTS1 and MCTS3 but can adjust its skill to them. AlphaDDA2 is stronger than MCTS4,

**Table 6 Win, loss, and draw rates of AlphaDDAs for Connect4 when they are the first player.**

| Opponent | AlphaDDA1 | | | AlphaDDA2 | | | AlphaDDA3 | | |
|---|---|---|---|---|---|---|---|---|---|
| | Win | Loss | Draw | Win | Loss | Draw | Win | Loss | Draw |
| AlphaZero | 0.02 | 0.04 | 0.94 | 0.02 | 0.24 | 0.74 | 0.0 | 0.44 | 0.56 |
| MCTS1 | 0.32 | 0.66 | 0.02 | 0.24 | 0.76 | 0.0 | 0.0 | 1.0 | 0.0 |
| MCTS2 | 0.54 | 0.46 | 0.0 | 0.58 | 0.4 | 0.02 | 0.02 | 0.96 | 0.02 |
| MCTS3 | 0.44 | 0.5 | 0.06 | 0.36 | 0.54 | 0.1 | 0.0 | 0.98 | 0.02 |
| MCTS4 | 0.76 | 0.22 | 0.02 | 0.78 | 0.22 | 0.0 | 0.04 | 0.94 | 0.02 |
| Minimax1 | 0.94 | 0.06 | 0.0 | 0.78 | 0.22 | 0.0 | 0.0 | 0.98 | 0.02 |
| Minimax2 | 0.88 | 0.12 | 0.0 | 0.94 | 0.06 | 0.0 | 0.08 | 0.92 | 0.0 |
| Random | 1.0 | 0.0 | 0.0 | 1.0 | 0.0 | 0.0 | 0.56 | 0.4 | 0.04 |

**Table 7 Win, loss, and draw rates of AlphaDDAs for Connect4 when they are the second player.**

| Opponent | AlphaDDA1 | | | AlphaDDA2 | | | AlphaDDA3 | | |
|---|---|---|---|---|---|---|---|---|---|
| | Win | Loss | Draw | Win | Loss | Draw | Win | Loss | Draw |
| AlphaZero | 0.02 | 0.0 | 0.98 | 0.0 | 0.14 | 0.86 | 0.0 | 1.0 | 0.0 |
| MCTS1 | 0.12 | 0.86 | 0.02 | 0.34 | 0.58 | 0.08 | 0.0 | 1.0 | 0.0 |
| MCTS2 | 0.38 | 0.62 | 0.0 | 0.52 | 0.44 | 0.04 | 0.0 | 1.0 | 0.0 |
| MCTS3 | 0.2 | 0.76 | 0.04 | 0.42 | 0.5 | 0.08 | 0.0 | 0.98 | 0.02 |
| MCTS4 | 0.4 | 0.6 | 0.0 | 0.8 | 0.18 | 0.02 | 0.0 | 0.96 | 0.04 |
| Minimax1 | 0.44 | 0.54 | 0.02 | 0.68 | 0.32 | 0.0 | 0.0 | 1.0 | 0.0 |
| Minimax2 | 0.48 | 0.52 | 0.0 | 0.94 | 0.06 | 0.0 | 0.0 | 1.0 | 0.0 |
| Random | 0.96 | 0.04 | 0.0 | 1.0 | 0.0 | 0.0 | 0.36 | 0.58 | 0.06 |

and Minimax1 but adjusts its skill to them. AlphaDDA2 defeats Minimax2 and Random. Thus, AlphaDDA2 can adjust its skill to AlphaZero, MCTS1, MCTS2, MCTS3, MCTS4, and Minimax1.

AlphaDDA3 is defeated by MCTS1, MCTS2, MCTS3, MCTS4, Minimax1, and Minimax2. AlphaDDA3 does not win against AlphaZero but has a chance to draw against AlphaZero. AlphaDDA3 is on par with Random. Thus, AlphaDDA3 can adjust its skill to Random.

Tables 6 and 7 show the win, loss, and draw rates of AlphaDDAs against the AI agents for Connect4 when AlphaDDAs are the first and the second players, respectively. These win-loss-draw rates are calculated through 50 games with each AI. AlphaDDA1 for the first player is stronger than for the second player. AlphaDDA1 for the first player defeats Minimax1. This result suggests that AlphaDDA1 for the first player does not adjust its skill to Minimax1. The win-loss-draw rates of AlphaDDA2 for the first player show a similar tendency to those for the second player. AlphaDDA3 for the first player draws against AlphaZero with a probability of about 0.5. However, AlphaDDA3 for the second player is defeated by AlphaZero. This result shows that AlphaDDA3 for the second player cannot adjust its strength to AlphaZero.

**Table 8 Win, loss, and draw rates of AlphaDDAs for 6x6 Othello.**

| Opponent | AlphaDDA1 | | | AlphaDDA2 | | | AlphaDDA3 | | |
|---|---|---|---|---|---|---|---|---|---|
| | Win | Loss | Draw | Win | Loss | Draw | Win | Loss | Draw |
| AlphaZero | 0.23 | 0.62 | 0.15 | 0.19 | 0.8 | 0.01 | 0.0 | 1.0 | 0.0 |
| MCTS1 | 0.29 | 0.61 | 0.1 | 0.27 | 0.67 | 0.06 | 0.01 | 0.98 | 0.01 |
| MCTS2 | 0.57 | 0.38 | 0.05 | 0.55 | 0.38 | 0.07 | 0.0 | 0.93 | 0.07 |
| MCTS3 | 0.4 | 0.54 | 0.06 | 0.52 | 0.44 | 0.04 | 0.02 | 0.98 | 0.0 |
| MCTS4 | 0.62 | 0.33 | 0.05 | 0.69 | 0.28 | 0.03 | 0.03 | 0.92 | 0.05 |
| Minimax1 | 0.65 | 0.34 | 0.01 | 0.75 | 0.19 | 0.06 | 0.27 | 0.44 | 0.29 |
| Minimax2 | 0.96 | 0.04 | 0.0 | 0.95 | 0.05 | 0.0 | 0.63 | 0.06 | 0.31 |
| Random | 0.96 | 0.02 | 0.02 | 0.94 | 0.04 | 0.02 | 0.4 | 0.44 | 0.16 |

## 6x6 Othello

Table 8 shows the win, loss, and draw rates of AlphaDDAs and the AI agents for 6x6 Othello. The win-loss-draw rate is calculated through 100 games with each AI (50 games when AlphaDDA is the first player and 50 games when AlphaDDA is the second player).

AlphaDDA1 is on par with MCTS3. AlphaDDA1 is weaker than AlphaZero and MCTS1 but can adjust its skill against the opponent. AlphaDDA1 is stronger than MCTS2, MCTS4, and Minimax1 but can adjust its skill against the opponents. AlphaDDA1 defeats Minimax2 and Random. These results show that AlphaDDA1 can adjust its skill to AlphaZero, MCTS1, MCTS2, MCTS3, MCTS4, and Minimax1.

AlphaDDA2 is on par with MCTS3. AlphaDDA2 is weaker than MCTS1 but adjusts its skill to the opponent. AlphaDDA2 is stronger than MCTS2, MCTS4, and Minimax1 but can adjust its skill to the opponents. AlphaDDA2 defeats Minimax2 and Random. These results suggest that AlphaDDA2 can adjust its skill to MCTS1, MCTS2, MCTS3, MCTS4, and Minimax1.

AlphaDDA3 is defeated by AlphaZero, MCTS1, MCTS2, MCTS3, and MCTS4. AlphaDDA3 is on par with Random. AlphaDDA3 is weaker than Minimax1 but can adjust its skill against the opponent. AlphaDDA3 is stronger than Minimax2 but has a chance to draw against the opponent. Thus, AlphaDDA3 adjusts its skill to Minimax1 and Random.

Tables 9 and 10 show the win, loss, and draw rates of AlphaDDAs against the AI agents for 6x6 Othello when AlphaDDAs are the first and the second players, respectively. These win-loss-draw rates are calculated through 50 games with each AI. AlphaDDA1 for the first player is slightly stronger than MCTS2. AlphaDDA1 for the first player defeats Minimax1. AlphaDDA1 for the second player loses against AlphaZero but draws against AlphaZero with a probability of 0.3. These results show that AlphaDDA1 for the first player does not adjust its skill to Minimax1. AlphaDDA2 for the second player adjust its skill to AlphaZero. AlphaDDA3 for the first player defeats against Minimax2. This result shows that AlphaDDA3 for the first player does not adjust its skill against Minimax2.

**Table 9 Win, loss, and draw rates of AlphaDDAs for 6x6 Othello when they are the first players.**

| Opponent | AlphaDDA1 | | | AlphaDDA2 | | | AlphaDDA3 | | |
|---|---|---|---|---|---|---|---|---|---|
| | Win | Loss | Draw | Win | Loss | Draw | Win | Loss | Draw |
| AlphaZero | 0.4 | 0.6 | 0.0 | 0.08 | 0.92 | 0.0 | 0.0 | 1.0 | 0.0 |
| MCTS1 | 0.3 | 0.62 | 0.08 | 0.32 | 0.62 | 0.06 | 0.0 | 0.98 | 0.02 |
| MCTS2 | 0.64 | 0.28 | 0.08 | 0.58 | 0.34 | 0.08 | 0.0 | 0.98 | 0.02 |
| MCTS3 | 0.48 | 0.46 | 0.06 | 0.42 | 0.54 | 0.04 | 0.0 | 1.0 | 0.0 |
| MCTS4 | 0.66 | 0.26 | 0.08 | 0.76 | 0.22 | 0.02 | 0.04 | 0.88 | 0.08 |
| Minimax1 | 0.88 | 0.1 | 0.02 | 0.74 | 0.22 | 0.04 | 0.18 | 0.26 | 0.56 |
| Minimax2 | 0.98 | 0.02 | 0.0 | 0.96 | 0.04 | 0.0 | 0.98 | 0.02 | 0.0 |
| Random | 0.96 | 0.04 | 0.0 | 0.92 | 0.04 | 0.04 | 0.3 | 0.54 | 0.16 |

**Table 10 Win, loss, and draw rates of AlphaDDAs for 6x6 Othello when they are the second players.**

| Opponent | AlphaDDA1 | | | AlphaDDA2 | | | AlphaDDA3 | | |
|---|---|---|---|---|---|---|---|---|---|
| | Win | Loss | Draw | Win | Loss | Draw | Win | Loss | Draw |
| AlphaZero | 0.06 | 0.64 | 0.3 | 0.3 | 0.68 | 0.02 | 0.0 | 1.0 | 0.0 |
| MCTS1 | 0.28 | 0.6 | 0.12 | 0.22 | 0.72 | 0.06 | 0.02 | 0.98 | 0.0 |
| MCTS2 | 0.5 | 0.48 | 0.02 | 0.52 | 0.42 | 0.06 | 0.0 | 0.88 | 0.12 |
| MCTS3 | 0.32 | 0.62 | 0.06 | 0.62 | 0.34 | 0.04 | 0.04 | 0.96 | 0.0 |
| MCTS4 | 0.58 | 0.4 | 0.02 | 0.62 | 0.34 | 0.04 | 0.02 | 0.96 | 0.02 |
| Minimax1 | 0.42 | 0.58 | 0.0 | 0.76 | 0.16 | 0.08 | 0.36 | 0.62 | 0.02 |
| Minimax2 | 0.94 | 0.06 | 0.0 | 0.94 | 0.06 | 0.0 | 0.28 | 0.1 | 0.62 |
| Random | 0.96 | 0.0 | 0.04 | 0.96 | 0.04 | 0.0 | 0.5 | 0.34 | 0.16 |

## Othello

Table 11 shows the win, loss, and draw rates of AlphaDDAs and the AI agents for Othello. The win-loss-draw rate is calculated through 100 games with each AI (50 games when AlphaDDA is the first player and 50 games when AlphaDDA is the second player).

AlphaDDA1 is on par with AlphaZero and MCTS2 because its win rates against them are 0.56 and 0.52, respectively. AlphaDDA1 is stronger than MCTS4, Minimax1, and Minimax2 but can adjust its skill to the opponents. AlphaDDA1 is weaker than MCTS1 and MCTS3 but can adjust its skill to the opponents. AlphaDDA1 defeats Random. These results suggest that AlphaDDA1 can adjust its skill to AlphaZero, MCTS1, MCTS2, MCTS3, MCTS4, Minimax1, and Minimax2.

AlphaDDA2 is on par with AlphaZero, and MCTS3. AlphaDDA2 is weaker than MCTS1 but can adjust its skill to MCTS1. AlphaDDA2 is stronger than MCTS2, MCTS4, and Minimax1 but can change its skill. AlphaDDA2 defeats Random. Thus, AlphaDDA2 can change its skill against AlphaZero, MCTS1, MCTS2, MCTS3, MCTS4, and Minimax1.

AlphaDDA3 is on par with Random. AlphaDDA3 is weaker than Minimax1 but has a chance of a draw. It is defeated by AlphaZero, MCTS1, MCTS2, MCTS3, and MCTS4. Thus, AlphaDDA3 can adjust its skill to Random.

**Table 11 Win, loss, and draw rates of AlphaDDAs for Othello.**

| Opponent | AlphaDDA1 | | | AlphaDDA2 | | | AlphaDDA3 | | |
|---|---|---|---|---|---|---|---|---|---|
| | Win | Loss | Draw | Win | Loss | Draw | Win | Loss | Draw |
| AlphaZero | 0.56 | 0.44 | 0.0 | 0.47 | 0.5 | 0.03 | 0.0 | 1.0 | 0.0 |
| MCTS1 | 0.35 | 0.6 | 0.05 | 0.35 | 0.62 | 0.03 | 0.0 | 0.96 | 0.04 |
| MCTS2 | 0.52 | 0.43 | 0.05 | 0.55 | 0.37 | 0.08 | 0.01 | 0.92 | 0.07 |
| MCTS3 | 0.36 | 0.56 | 0.08 | 0.46 | 0.45 | 0.09 | 0.01 | 0.98 | 0.01 |
| MCTS4 | 0.71 | 0.27 | 0.02 | 0.73 | 0.21 | 0.06 | 0.05 | 0.89 | 0.06 |
| Minimax1 | 0.64 | 0.32 | 0.04 | 0.74 | 0.22 | 0.04 | 0.13 | 0.67 | 0.2 |
| Minimax2 | 0.59 | 0.39 | 0.02 | 0.82 | 0.17 | 0.01 | 0.13 | 0.87 | 0.0 |
| Random | 0.96 | 0.03 | 0.01 | 0.99 | 0.01 | 0.0 | 0.43 | 0.38 | 0.19 |

**Table 12 Win, loss, and draw rates of AlphaDDAs for Othello when they are the first players.**

| Opponent | AlphaDDA1 | | | AlphaDDA2 | | | AlphaDDA3 | | |
|---|---|---|---|---|---|---|---|---|---|
| | Win | Loss | Draw | Win | Loss | Draw | Win | Loss | Draw |
| AlphaZero | 0.52 | 0.48 | 0.0 | 0.38 | 0.6 | 0.02 | 0.0 | 1.0 | 0.0 |
| MCTS1 | 0.36 | 0.6 | 0.04 | 0.36 | 0.64 | 0.0 | 0.0 | 0.96 | 0.04 |
| MCTS2 | 0.52 | 0.42 | 0.06 | 0.56 | 0.42 | 0.02 | 0.02 | 0.9 | 0.08 |
| MCTS3 | 0.44 | 0.5 | 0.06 | 0.4 | 0.48 | 0.12 | 0.0 | 0.98 | 0.02 |
| MCTS4 | 0.8 | 0.18 | 0.02 | 0.62 | 0.34 | 0.04 | 0.04 | 0.9 | 0.06 |
| Minimax1 | 0.58 | 0.4 | 0.02 | 0.72 | 0.22 | 0.06 | 0.1 | 0.84 | 0.06 |
| Minimax2 | 0.9 | 0.08 | 0.02 | 0.82 | 0.16 | 0.02 | 0.06 | 0.94 | 0.0 |
| Random | 0.98 | 0.0 | 0.02 | 1.0 | 0.0 | 0.0 | 0.56 | 0.28 | 0.16 |

**Table 13 Win, loss, and draw rates of AlphaDDAs for Othello when they are the second players.**

| Opponent | AlphaDDA1 | | | AlphaDDA2 | | | AlphaDDA3 | | |
|---|---|---|---|---|---|---|---|---|---|
| | Win | Loss | Draw | Win | Loss | Draw | Win | Loss | Draw |
| AlphaZero | 0.6 | 0.4 | 0.0 | 0.56 | 0.4 | 0.04 | 0.0 | 1.0 | 0.0 |
| MCTS1 | 0.34 | 0.6 | 0.06 | 0.34 | 0.6 | 0.06 | 0.0 | 0.96 | 0.04 |
| MCTS2 | 0.52 | 0.44 | 0.04 | 0.54 | 0.32 | 0.14 | 0.0 | 0.94 | 0.06 |
| MCTS3 | 0.28 | 0.62 | 0.1 | 0.52 | 0.42 | 0.06 | 0.02 | 0.98 | 0.0 |
| MCTS4 | 0.62 | 0.36 | 0.02 | 0.84 | 0.08 | 0.08 | 0.06 | 0.88 | 0.06 |
| Minimax1 | 0.7 | 0.24 | 0.06 | 0.76 | 0.22 | 0.02 | 0.16 | 0.5 | 0.34 |
| Minimax2 | 0.28 | 0.7 | 0.02 | 0.82 | 0.18 | 0.0 | 0.2 | 0.8 | 0.0 |
| Random | 0.94 | 0.06 | 0.0 | 0.98 | 0.02 | 0.0 | 0.3 | 0.48 | 0.22 |

Tables 12 and 13 show the win, loss, and draw rates of AlphaDDAs against the AI agents for Othello when AlphaDDAs are the first and the second players, respectively. The win-loss-draw rate is calculated through 50 games with each AI. AlphaDDA1 for the first player is stronger than MCTS4. AlphaDDA1 for the second player is stronger than Minimax1. AlphaDDA1 for the second player is slightly weaker than MCTS3. AlphaDDA1 for the first

player defeats Minimax2. This result suggests that AlphaDDA1 for the first player does not adjust its skill to Minimax2. AlphaDDA2 for the second player defeats MCTS4. This result suggests that AlphaDDA2 for the second player does not adjust its skill to MCTS4. AlphaDDA3 for the first player is slightly stronger than Random.

## CONCLUSION AND DISCUSSION

This study aimed to develop AlphaDDA, a game-playing AI that can change its playing skill according to the state of the game. AlphaDDA estimates the state's value and the move probability from the board state using the DNN and decides a move using MCTS, as in AlphaZero. In this study, three types of AlphaDDA are proposed: AlphaDDA1, AlphaDDA2, and AlphaDDA3. AlphaDDA1 and AlphaDDA2 adapt their skill to the opponent by changing the number of simulations and dropout probability according to the value, respectively. AlphaDDA3 decides the next move by MCTS using a new UCT score. The DDA performance of the AlphaDDAs was evaluated for Connect4, 6x6 Othello, and Othello. The AlphaDDAs played these games with AlphaZero, MCTS1, MCTS2, MCTS3, MCTS4, Minimax1, Minimax2, and Random. The results of this evaluation show that AlphaDDA1 and AlphaDDA2 adjust their skills to AlphaZero, MCTSs, and Minimaxs. However, AlphaDDA1 and AlphaDDA2 defeat the weakest opponent Random. Furthermore, the results show that AlphaDDA3 can adjust its skill against Random. However, AlphaDDA3 is beaten by the other AI agents, excluding Random. Thus, the approaches of AlphaDDA1 and AlphaDDA2 are practical for DDA.

The AlphaDDA approach for DDA is very simple. The approach is to balance AI's skill by changing its parameter according to the value estimated by the DNN. If the DNN can accurately estimate the value of a game state and we know a parameter dominantly controlling AI's skills, an AI system can change its skills according to the game state. In other words, we can embed the approach to any algorithm as long as the DNN can accurately evaluate a game state and we know the dominant parameter. For example, the approach can be applied to the vanilla MCTS and minimax algorithm if the DNN can estimate the value of a game state. The vanilla MCTS with DDA changes the number of simulations according to the value estimated by the DNN, such as AlphaDDA1. The minimax algorithm with DDA changes the search depth or evaluation function according to the value estimated by the DNN. Of course, we can embed the DDA approach to AlphaZero-based AI algorithms, such as AlphaZero in continuous action space (*Moerland et al., 2018*), multiplayer AlphaZero *Petosa & Balch (2019)*, and MuZero (*Schrittwieser et al., 2020*). However, it is difficult to use the DDA approach if the DNN cannot precisely estimate the value of a game state or we do not know a parameter critically controlling an AI agent's strength.

AlphaDDA1 and AlphaDDA2 cannot adjust their skills to Random. They cannot become weaker than the trained AlphaZero for $N_{sim} = 1$ and the trained AlphaZero with dropout for $P_{drop} = 0.95$, respectively. The trained AlphaZero for $N_{sim} = 1$ and the trained AlphaZero with dropout for $P_{drop} = 0.95$ are stronger than Random, as shown in Figs. 3A and 4. Thus, AlphaDDA1 and AlphaDDA2 cannot become weaker than Random.

AlphaDDA3 is defeated by AlphaZero, MCTSs, and Minimaxs. The purpose of AlphaDDA3's algorithm causes its weakness. The purpose is to balance the mean estimated value, not to win. The examples of games in "Examples of games" show that AlphaDDA3 can balance the mean estimated value and avoid its winning. AlphaDDA3 can avoid winning to balance the mean estimated value, but the avoidance makes the board worse for itself. In an endgame, it is difficult for AlphaDDA3 to improve the situation worsened by the avoidance of winning, and AlphaDDA3 will consequently not win, even though AlphaDDA3 chooses a better move to balance the mean estimated value.

The MCTS for AlphaDDA3 assumes that the opponent selects a move that renders the board state with the same value as the current board state. Furthermore, AlphaDDA3 selects the move rendering the board state with the inverse value as the current board state in the MCTS. For instance, when Random is an opponent, AlphaDDA3 selects the worse move because Random rarely selects a better move. It is difficult for AlphaDDA3 to adjust its skill to MCTS1, MCTS2, and Minimax because they may decide on a relatively better move but not the best one. In addition, AlphaDDA3 cannot beat AlphaZero. AlphaZero always selects the best move, and the average value becomes close to the disc color of AlphaZero. In this case, AlphaZero is extremely predominant, and it is difficult for AlphaDDA3 to beat AlphaZero even though AlphaDDA3 selects a better move.

We can improve the performance of AlphaDDAs. AlphaDDA1 and AlphaDDA2 have good DDA ability but defeat a very weak player that selects a move at random. The reason is that the algorithm based on AlphaZero cannot become as weak as an overly weak opponent such as Random. One approach to improve this issue is to change the AlphaZero-based AI algorithm to a weaker one when an opponent is too weak. There are two methods to achieve this approach. One method is that the AI with DDA evaluates the opponent's skill using the opponent's game history and changes the AI algorithm. The other method is that the AI with DDA changes the game AI algorithm if the board state continues to be worse for the opponent during the current game.

# APPENDIX

## Alphazero

AlphaDDA is the trained AlphaZero with an added DDA. AlphaZero consists of the DNN and MCTS, as shown in Fig. 1A. The DNN receives an input consisting of the board states and the disc color of the current player. The DNN provides the value of a state and the move probabilities that indicate the probabilities of selecting valid moves. The MCTS is used to decide the best move based on the value and move probabilities.

### Deep neural network in AlphaZero

The DNN predicts the value $v(s)$ and the move probability $\boldsymbol{p}$ with components $p(a|s)$ for each $a$, where $s$ is a state and $a$ is an action. In a board game, $s$ and $a$ denote a board state and a move, respectively. The input of the DNN consists of the board state and the disc color of the current player. The architecture of the DNN is shown in Fig. A1. The DNN has a "Body" that consists of residual blocks and a "Head," which consists of the value head and the policy head. The value head and the policy head output the value $v(s)$
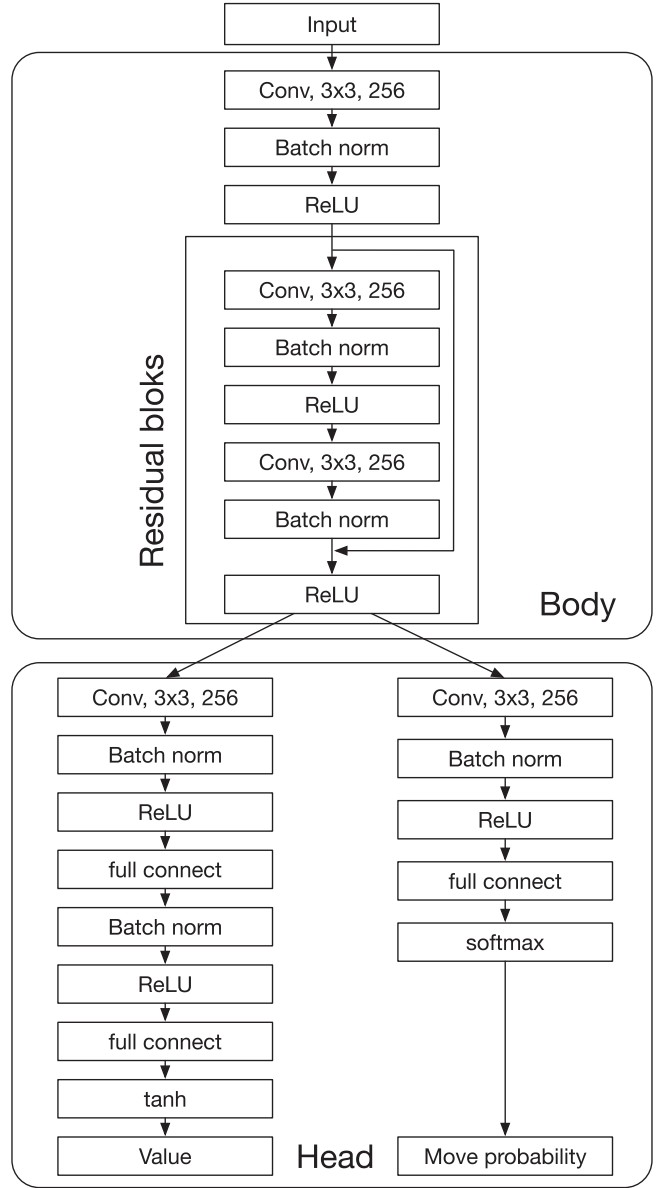

**Figure A1 Deep neural network of AlphaDDA.**

and the move probabilities $p$, respectively. AlphaDDA's DNN is the same as that of AlphaZero.

The input to the DNN is an $M \times N \times (2T + 1)$ image stack that consists of $2T + 1$ binary feature planes of size $M \times N$, where $M \times N$ is the board size, and $T$ is the number of histories. The first $T$ feature planes represent the occupation of the discs of one player. A feature value is 1 if the corresponding cell is occupied by the disc, and 0 otherwise. The next $T$ feature planes similarly represent the occupation of the other player's discs. The final feature plane represents the disc color of the current player. The store colors of the first and second players are represented by 1 and −1, respectively.

### Monte Carlo tree search in AlphaZero

This subsection explains the Monte Carlo tree search (MCTS) in AlphaZero. Each node in the game tree represents the state of the game and has an edge $(s, a)$ for all valid actions. Each edge stores a set of statistics, $\{N(s, a), W(s, a), Q(s, a), P(s, a)\}$, where $N(s, a)$ is the visit count, $W(s, a)$ is the cumulative value, $Q(s, a)$ is the mean value $Q(s, a) = W(s, a)/N(s, a)$, and $P(s, a)$ is the move probability. The MCTS for AlphaZero consists of four steps: *Select*, *Expand and Evaluate*, *Backup*, and *Play*. The set of three steps, *Select*, *Expand and Evaluate*, and *Backup*, is called simulation and is repeated $N_{sim}$ times. *Play* is performed after $N_{sim}$ simulations.

In *Select*, the tree is traveled from the root node $s_{root}$ to the leaf node $s_L$ at time step $L$ using a variant of the polynomial upper confidence bound for tree search (PUCT) algorithm. At each time step $t < L$, the selected action $a_t$ has the maximum score, as described by the following equation:

$$a_t = \arg\max_a \left( Q(s_t, a) + C_{puct} P(s_t, a) \frac{\sqrt{N(s_t)}}{1 + N(s_t, a)} \right), \tag{8}$$

where $N(s_t)$ is the parent visit count, and $C_{puct}$ is the exploration rate. In this study, $C_{puct}$ is constant, but in the original AlphaZero $C_{puct}$ slowly increases with the search time.

In *Expand and Evaluate*, the DNN evaluates the leaf node and outputs $v(s_l)$ and $\boldsymbol{p}_a(s_l)$. If the leaf node is a terminal node, $v(s_L)$ is the winning player's disc color. The leaf node is expanded, and each edge $(s_L, a)$ is initialized to $\{N(s_L, a) = 0, W(s_L, a) = 0, Q(s_L, a) = 0, P(s_L, a) = p_a\}$.

In *Backup*, the visit counts and values are updated in a backward pass through each step $t \leq L$. The visit count is incremented by 1, $N(s_t, a_t) \leftarrow N(s_t, a_t) + 1$, and the cumulative value and the mean value are updated, $W(s_t, a_t) \leftarrow W(s_t, a_t) + v$, $Q(s_t, a) \leftarrow W(s_t, a_t)/N(s_t, a_t)$.

In *Play*, AlphaZero selects the action corresponding to the most visited edge from the root node.

### Training

In training, the set of three processes, *Self-play*, *Augmentation*, and *Learning*, are iterated $N_{iter}$ times. The training algorithm was slightly improved from the original AlphaZero to train the DNN using a single computer.

In *Self-play*, AlphaZero plays the game with itself $N_{self}$ times. Before $T_{opening}$ turns, AlphaZero stochastically selects the next action from valid moves based on the softmax function:

$$p(a|s) = \exp(N(s, a)/\tau)/\Sigma_b \exp(N(s, b)/\tau), \tag{9}$$

where $\tau$ is a temperature parameter that controls the level of exploration. Otherwise, the maximum visited action is selected. Through the stochastic decision, AlphaZero obtains the possibility of learning a new and better action. Through *Self-play*, we obtain the board states, winner, and search probabilities. The winner and the search probabilities are

**Table A1 Parameters.**

| Parameter | Connect4 | 6x6 Othello | Othello |
|---|---|---|---|
| $N_{iter}$ | 600 | 600 | 700 |
| $N_{self}$ | 30 | 10 | 10 |
| $N_{sim}$ | 200 | 200 | 400 |
| $C_{puct}$ | 1.25 | 1.25 | 1.25 |
| $T_{opening}$ | 4 | 4 | 6 |
| $\tau$ | 50 | 20 | 40 |
| $\varepsilon$ | 0.2 | 0.2 | 0.2 |
| $T$ | 1 | 1 | 1 |
| Number of residual blocks | 3 | 3 | 5 |
| Kernel size | 3 | 3 | 3 |
| Number of filters | 256 | 256 | 256 |
| $N_{queue}$ | 20,000 | 20,000 | 50,000 |
| $N_{epoch}$ | 1 | 1 | 1 |
| $N_{batch}$ | 2,048 | 2,048 | 2,048 |
| Learning rate | 0.2 | 0.2 | 0.2 |
| Momentum | 0.9 | 0.9 | 0.9 |
| Weight decay | 0.0001 | 0.0001 | 0.0001 |

saved for each board state. The search probabilities are the probabilities of selecting valid moves at the root node in the MCTS.

In *Augmentation*, the data obtained in *Self-play* are augmented by generating two and eight symmetries for each position for Connect4 and the Othellos, respectively. The augmented board states are added to the queue in which $N_{queue}$ board states can be stored. This queue is the training dataset.

In *Learning*, the DNN is trained using the mini-batch method with $N_{batch}$ batches and $N_{epochs}$ epochs. The stochastic gradient descent with momentum and weight decay is applied for the training. The loss function $l$ sums the mean-squared error between the disc color of the winner and the value, and the cross-entropy losses between the search probabilities and the move probability.

$$l = (c_{win} - v)^2 - \pi^{\mathrm{T}} \log p, \tag{10}$$

where $c_{win}$ is the disc color of the winner, $v$ is the value estimated by the DNN, $\pi$ represents the search probabilities, and $p$ is the move probability.

### Parameters

The parameters of AlphaZero are listed in Table A1. The weight decay and momentum are the same as those in the pseudocode of AlphaZero (*Silver et al., 2018*). The kernel size and number of channels are the same as those of the original AlphaZero. The learning rate and $C_{puct}$ are the same as the initial values of the original AlphaZero. The other parameters are hand-tuned to maximize the strength of AlphaZero and minimize its learning time.

**Table A2 Sets of parameters for AlphaDDA1.**

| Game | $N_h$ | $A_{sim}$ | $B_{sim0}$ | $N_{max}$ |
|---|---|---|---|---|
| Connect4 | 1, 2, 3, 4 | 1.6, 2.0, 2.4 | $-1.6, -1.5, -1.4, -1.3, -1.2$ | 200, 300, 400 |
| 6x6 Othello | 1, 2, 3, 4, 5 | 1.4, 1.6, 2.0, 2.4 | $-1.8, -1.6, -1.4, -1.2$ | 200, 300, 400 |
| Othello | 1, 2, 3, 4, 5 | 1.6, 2.0, 2.4, 2.8 | $-1.8, -1.6, -1.4, -1.2$ | 400, 600 |

**Table A3 Sets of parameters for AlphaDDA2.**

| Game | $N_h$ | $A_{drop}$ | $P_{drop0}$ |
|---|---|---|---|
| Connect4 | 1, 2, 3, 4 | 0.5, 1, 5, 10, 20 | $-0.9, -0.8, -0.6, -0.4, -0.2, 0$ |
| 6x6 Othello | 1, 2, 3, 4 | 1, 5, 10, 20 | $-0.9, -0.8, -0.6, -0.5, -0.4$ |
| Othello | 1, 2, 3, 4, 5 | 5, 10, 20, 40 | $-0.9, -0.8, -0.6, -0.4$ |

**Table A4 Sets of parameters for AlphaDDA3.**

| Game | $N_h$ | $C$ |
|---|---|---|
| Connect4 | 1, 2, 3, 4, 5 | 0.25, 0.5, 0.75, 1.0, 1.25 |
| 6x6 Othello | 1, 2, 3, 4, 5 | 0.25, 0.5, 0.6, 0.75, 0.9, 1.0 |
| Othello | 1, 2, 3, 4, 5 | 0.25, 0.5, 0.6, 0.75, 0.9, 1.0 |

## Grid search

In this study, grid search is used for hyperparameter tuning. Grid search is the most basic hyperparameter optimization method (*Feurer & Hutter, 2019*) and is simple to implement (*Bergstra & Bengio, 2012*).

In grid search, the sets of the parameters for AlphaDDA1, AlphaDDA2, and AlphaDDA3 shown in Tables A2–A4 are evaluated. AlphaDDAs play the games with AlphaZero, MCTS1, MCTS2, Minimax1, and Random. AlphaDDAs with each pair of parameters play forty games with each AI (20 games when AlphaDDA is the first player and 20 games when AlphaDDA is the second player). Grid search finds the set of the parameters that achieves the smallest performance metric.

The performance metric is the sum of the differences between the number of wins and the number of losses. The following equation denotes the mathematical description of the metric $D$.

$$D = \sum_{X} (|N^{Win}_{X_{first}} - N^{Loss}_{X_{first}}| + |N^{Win}_{X_{second}} - N^{Loss}_{X_{second}}|),$$

$$X = \{AlphaZero, MCTS1, MCTS2, Minimax1, Random\}.$$

(11)

$N^{Win}_{X_{first}}$ and $N^{Loss}_{X_{first}}$ are the number of wins and the number of losses, respectively, when the opponent $X$ is the first player. $N^{Win}_{X_{second}}$ and $N^{Loss}_{X_{second}}$ are the number of wins and the number of losses, respectively, when the opponent $X$ the second player.

## Monte carlo tree search

The MCTS is a best-first search method that does not require an evaluation function (*Winands, 2017*). MCTS consists of four strategic steps: *Selection step*, *Playout step*, *Expansion step*, and *Backpropagation step* (*Winands, 2017*). In the *Selection step*, the tree is traversed from the root node to a leaf node. The child node $c$ of the parent node $p$ is selected to maximize the following score:

$$\text{UCT} = q_c/N_c + C\sqrt{\frac{\ln(N_p + 1)}{N_c + \varepsilon}}, \tag{12}$$

where $q_c$ is the cumulative value of $c$, $N_p$ is the visit count of $p$, $N_c$ is the visit count of $c$, and $\varepsilon$ is a constant value to avoid division by zero. In the *Playout step*, a valid move is selected at random until the end of the game is reached. In the *Expansion step*, child nodes are added to the leaf node when the visit count of the leaf node reaches $N_{\text{open}}$. The child nodes correspond to the valid moves. In the *Backpropagation step*, the result of the playout is propagated along the path from the leaf node to the root node. If the MCTS player itself wins, loses, and draws, $q_i \leftarrow q_i + 1$, $q_i \leftarrow q_i - 1$, and $q_i \leftarrow q_i$, respectively. $q_i$ is the cumulative value of node $i$. The set of the four steps mentioned above is called simulation. The simulation is executed $N_{\text{sim}}$ times. MCTS selects the action corresponding to the most visited node from the root node. In this study, $C = 0.5$, $\varepsilon = 10^{-7}$, and $N_{\text{open}} = 5$.

## Minimax algorithm

The minimax algorithm is a basic game tree search that finds the action generating the best value. Each node in the tree has a state, player, action, and value. The minimax algorithm first creates a game tree with a depth of $N_{\text{depth}}$. In this study, $N_{\text{depth}} = 3$. Here, the root node corresponds to the current state and the minimax player itself. Second, the states corresponding to the leaf nodes are evaluated. Third, all nodes are evaluated from the parents of the leaf nodes to the children of the root node. If the player corresponding to the node is the opponent, the value of the node is the minimum value of its child nodes. Otherwise, the value of the node is the maximum value of its child nodes. Finally, the minimax algorithm selects the action corresponding to the child node of the root with the maximum value.

The evaluation of a leaf node is specialized for each game. For Connect4, the values of the connections of two same-color discs and three same-color discs are, respectively, $R \times c_{\text{disc}}c_{\text{minimax}}$ and $R^2 \times c_{\text{disc}}c_{\text{minimax}}$, where $c_{\text{disc}}$ and $c_{\text{minimax}}$ are the colors of the connecting discs and the minimax player's disc, respectively. The value of the node is the sum of the values of all the connections on the board corresponding to the node. The value of the terminal node is $R^3 c_{\text{win}}c_{\text{minimax}}$, where $c_{\text{win}}$ is the color of the winner. $R$ is 100 and 2 for Minimax1 and Minimax2, respectively.

For 6x6 Othello and Othello, the value of a node is calculated using the following equation:

$$E = \sum_x \sum_y v(x, y) o(x, y) c_{\text{minimax}},$$ (13)

where $v(x, y)$ is the value of the cell $(x, y)$, and $o(x, y)$ is the occupation of the cell $(x, y)$. For 6x6 Othello and Othello, Minimax1 evaluates each cell using Eqs. (14) and (15), respectively. For 6x6 Othello and Othello, Minimax2 evaluates each cell to 1. $o(x, y)$ is 1, $-1$, and 0 if the cell $(x, y)$ is occupied by the first player's disc, the second player's disc, and empty, respectively. For 6x6 Othello and Othello, the minimax algorithm expands the tree to the terminal nodes after the last six turns. The value of the terminal node is $E_{\text{end}} = 1000 c_{\text{win}} c_{\text{minimax}}$, where $c_{\text{win}}$ is the color of the winner.

$$v_{6\times6} = \begin{pmatrix} 30 & -5 & 2 & 2 & -5 & 30 \\ -5 & -15 & 3 & 3 & -15 & -5 \\ 2 & 3 & 0 & 0 & 3 & 2 \\ 2 & 3 & 0 & 0 & 3 & 2 \\ -5 & -15 & 3 & 3 & -15 & -5 \\ 30 & -5 & 2 & 2 & -5 & 30 \end{pmatrix}.$$ (14)

$$v_{8\times8} = \begin{pmatrix} 120 & -20 & 20 & 5 & 5 & 20 & -20 & 120 \\ -20 & -40 & -5 & -5 & -5 & -5 & -40 & -20 \\ 20 & -5 & 15 & 3 & 3 & 15 & -5 & 20 \\ 5 & -5 & 3 & 3 & 3 & 3 & -5 & 5 \\ 5 & -5 & 3 & 3 & 3 & 3 & -5 & 5 \\ 20 & -5 & 15 & 3 & 3 & 15 & -5 & 20 \\ -20 & -40 & -5 & -5 & -5 & -5 & -40 & -20 \\ 120 & -20 & 20 & 5 & 5 & 20 & -20 & 120 \end{pmatrix}.$$ (15)

## Elo rating

Elo rating is a popular metric to measure the relative strength of a player. We can estimate the probability that player A defeats player B $p(A \text{ defeats B})$ using the Elo rating $e$. $p(A \text{ defeats B})$ is denoted by

$$p(A \text{ defeats B}) = 1/(1 + 10^{(e(B)-e(A))/400}),$$ (16)

where $e(A)$ is the Elo rating of player A. After $N_G$ games, the new Elo rating of player A $e'(A)$ is calculated using the following equation:

$$e'(A) = e(A) + K(N_{\text{win}} - N_G \times p(A \text{ defeats B})),$$ (17)

where $N_{\text{win}}$ is the number of times that player A wins. In this study, $K = 8$.

## Examples of games

Figure A2 is an example of a game of MCTS2 *vs* AlphaDDA1 in Othello. When an estimated value is closer to $-1$, AlphaDDA1 is more confident of winning because it is the

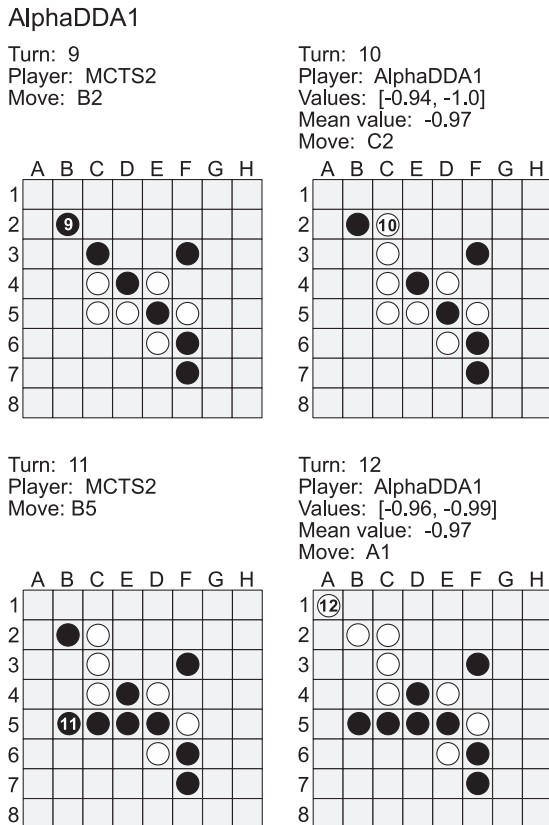

**Figure A2  Example of a game of MCTS2 (black) *vs* AlphaDDA1 (white) in Othello.**

second player in this case. MCTS2 played a move to the *B2* square at the 9th turn. This move is worse because the move increases the probability of AlphaDDA1 obtaining the *A1* corner. Corner discs are crucial because they cannot be flipped and commonly form an anchor which can stabilize other discs (*Rosenbloom, 1982*). In order to obtain the *A1* corner, AlphaDDA1 played the *C2* square at the 10th turn. MCTS1 could not flip the disc on *C3* at the 11th turn, and AlphaDDA1 put the disc on the *A1* square. Furthermore, AlphaDDA1 maintained the mean estimated value close to its disc color. This example suggests that AlphaDDA1 chooses a better move with a high estimated value.

Figure A3 is an example of a game of MCTS1 *vs* AlphaDDA1 in Connect4. MCTS1 dropped its disc into the *7* column at the 11th turn. MCTS1 will win if it drops its disc into the *6* column at its next turn. However, AlphaDDA1 dropped its disc into *1*. As a result, MCTS1 won. If AlphaDDA1 had searched the game tree to two depths, it could have avoided this loss. AlphaDDA1's move was caused by MCTS with a few simulations. If the mean estimated value is about −1 (AlphaDDA1 is the second player), the number of simulations of AlphaDDA1's MCTS is 7 in Connect4. AlphaDDA1 with a few simulations will choose a wrong move and not avoid losing not only because AlphaDDA1's MCTS shallowly searches the game tree but also because its DNN evaluates a situation on the assumption that its MCTS with sufficient simulations can avoid losing after a few turns.

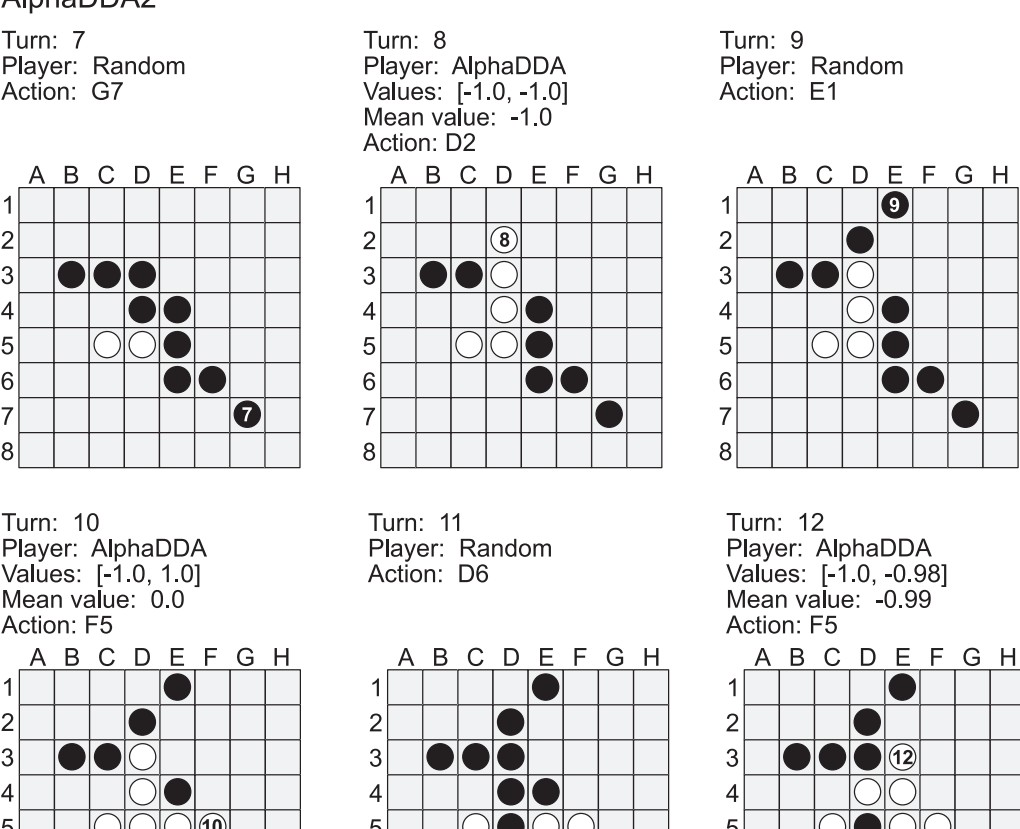

**AlphaDDA1**

**Figure A3** **Example of a game of MCTS1 (white) *vs* AlphaDDA1 (black) in Connect4.**

**AlphaDDA2**

**Figure A4** **Example of a game of Random (black) *vs* AlphaDDA2 (white) in Othello.**

This example is extreme, but it suggests that AlphaDDA1 may choose a wrong move when it is confident of winning.

Figure A4 is an example of a game of Random *vs* AlphaDDA2 in Othello. Random played a move to the *G7* square at the 7th turn. This is a worse move because the *G7* square

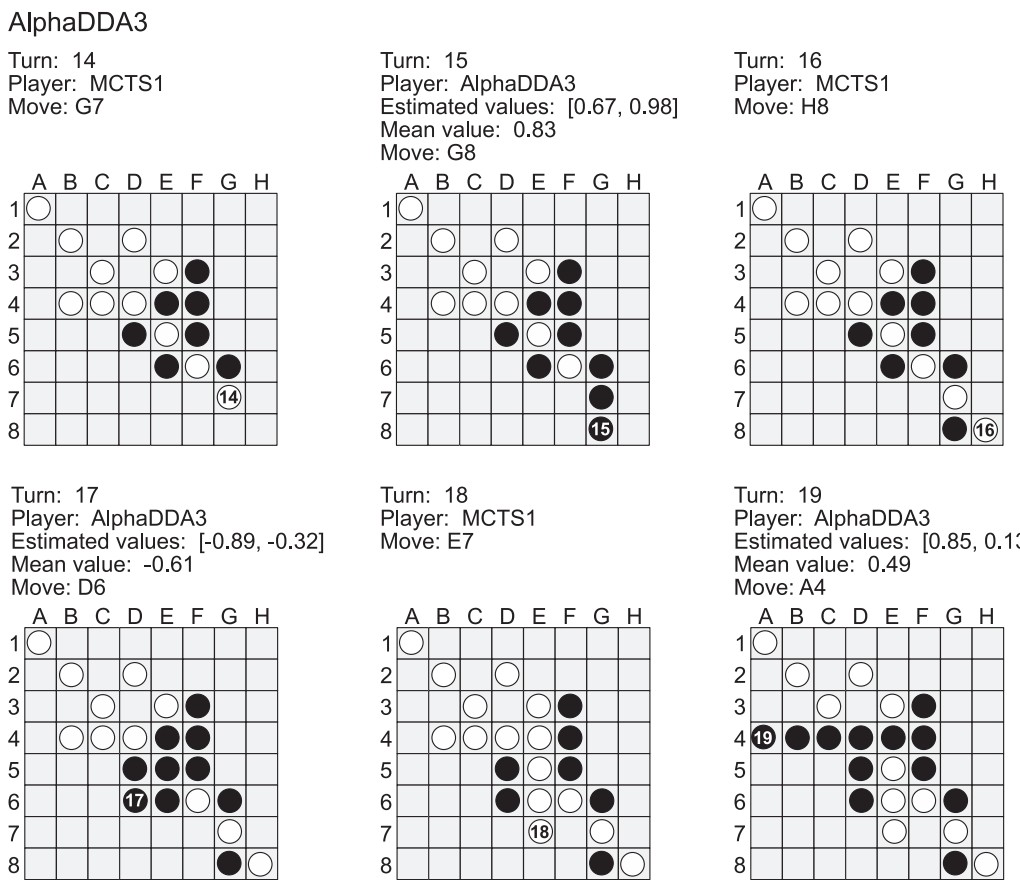

**AlphaDDA3**

**Figure A5** Example of a game of AlphaDDA3 (black) *vs* MCTS1 (white) in a Othello.

is a neighbor to the *H8* corner and this move allows AlphaDDA2 to obtain the corner. AlphaDDA2 played a move to the *D2* square at the 8th turn to obtain the corner. However, this move is not best because Random can obstruct AlphaDDA2's occupation of the corner square if Random puts its disc on the *C4*. The best move is *F5*. Random did not avoid the occupation of the corner at the 9th and the 11th turns. As a result, AlphaDDA2 could put its disc on the *H8* square at the 10th and the 12th turns. However, AlphaDDA2 did not put its disc on the *H8* square at the 10th and the 12th turns. These results show that AlphaDDA2 cannot choose the best move due to the deterioration of DNN by dropout.

Figure A5 is an example of a game of MCTS1 *vs* AlphaDDA3 in Othello. When an estimated value is closer to 1, AlphaDDA3 is more confident of winning because it is the first player in this game. MCTS1 played a move to the *G7* square at the 14th turn. This move is worse because AlphaDDA3 can occupy the *H8* corner square at the 17th turn if AlphaDDA3 puts a disc on *F7* at the 15 turn. However, AlphaDDA3 played *G8* at the 15th turn. This move worsens the situation for AlphaDDA3 because MCTS1 can put its disc on the *H8* corner at the next turn. In fact, this move changed the mean estimated value from 0.83 to −0.61. AlphaDDA3 was confident of winning at the 15th turn and balanced

AlphaDDA3 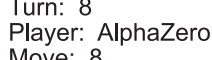

Turn: 8
Player: AlphaZero
Move: 8

Turn: 9
Player: AlphaDDA3
Values: [-0.69, -0.79]
Mean value: -0.74
Move: 5

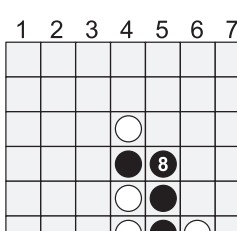

**Figure A6 Example of a game of AlphaDDA3 (white) *vs* AlphaZero (black) in a Connect4.**

the mean estimated value by choosing the bad move. On the other hand, AlphaDDA3 chose the move to improve the mean estimated value at the 17th turn, and the move changed the mean estimated value to 0.49 at the 19th turn. These results show that AlphaDDA3 can balance the estimated values, but the balancing accompanies a rapid change of the mean estimated value.

Figure A6 is an example of a game of AlphaDDA3 *vs* AlphaZero in Connect4. AlphaDDA3 is the first player in this game, and its disc color is 1 (white). AlphaZero dropped a disc into the *5* column at the 8th turn. This move resulted in three black discs in a column. AlphaDDA3 predicted its loss and avoided the loss by playing *5*. This result shows that AlphaDDA3 can avoid its loss when AlphaDDA3 predicts its loss. In other words, AlphaDDA3 may not avoid its loss when it predicts its win.

## Funding
The author received no funding for this work.

## Competing Interests
The author declares that they have no competing interests.

## Author Contributions
- Kazuhisa Fujita conceived and designed the experiments, performed the experiments, analyzed the data, performed the computation work, prepared figures and/or tables, authored or reviewed drafts of the article, and approved the final draft.

## Data Availability
The source code is available at GitHub: https://github.com/KazuhisaFujita/AlphaDDA.

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
