# Peer review of "AlphaDDA: strategies for adjusting the playing strength of a fully trained AlphaZero system to a suitable human training partner"

_PeerJ Computer Science, doi:10.7717/peerj-cs.1123_

## Round 0.1 · original submission · Major Revisions

A major revision is needed to improve the paper and address the criticisms raised. Please provide a detailed response letter. Thanks.

Reviewer 1 ·

Basic reporting

The paper addressed a very interesting problem that current game AIs, especially DNN based ones, are too much powerful that even human experts can not get win, making games are not attractive to human players. So this Paper proposes a so-called AlphaDDA method to enable a game-playing AI change its playing skill according to the state of the game based on AlphaZero. Three types of AlphaDDA are proposed, namely AlphaDDA1, AlphaDDA2, and AlphaDDA3.
It claims that AlphaDDA1 and AlphaDDA2 can adjust their skills to AlphaZero, MCTS1, MCTS2, and Minimax. And AlphaDDA3 can adjust its skill against Random. The paper is well written.
However, there are several concerns that must be addressed.

1. AlphaDDA1 and AlphaDDA2 are the almost the same as AlphaZero with a little changes on simulation or dropout. The values(table 1, 2) that chose for determining the simulation and dropout are based on a pre-search/tuning process by minimizing the difference of win or loss between DDA and other AIs, then why not tune different simulation or dropout values directly to minimize this difference? Even though, will the different AIs make these values(in table 1,2) different? A more complete grid search is needed.
2. AlphaDDA3 performs like a random player when looking at the win or loss with different AIs, especially for Connect4 why? Does the designed new UCT really make sense and work?
3. For the results in table 5,6,7, please provide the detail win or loss results with different hands(For example, in Table 7, AlphaZeroADD1 wins 50% and losses 50% against to AlphaZero, but it may indicate AlphaZeroADD1 wins at the first hand and losses the second hand always. This is import for not making human player get bored. A draw in this case is not the initial aim.)
4. Following qeustion 1 and 3, now that the value settings in table 1,2 are based on pre-tunings to minimize the difference of win or loss between AlphaZeroADDs with different other AIs, then the results for ADD1 and ADD2 are as that it should be. This is quite risky for the logic of this paper. Please provide convincing explanation.
5. To validate question 4, please provide the detail differences of grid-search for minimizing win or loss between AlphaZeroADD1,2,3 with other AIs respectively. If AlphaZeroADD1,2 minimized to a lower difference, but AlphaZeroADD3 has much higher difference(like showed in tables 5,6,7), then the logic is more likely wrong. Otherwise, there must be some other reasons/explanations.

Experimental design

no comment, see basic reporting

Validity of the findings

no comment, see basic reporting

Additional comments

no comment, see basic reporting

Reviewer 2 ·

Basic reporting

In the present manuscript, the authors propose AlphaDDA, an AlphaZero-based AI with dynamic difficulty adjustment (DDA). AlphaDDA can adjust its skill using only the state of a game without any prior knowledge regarding an opponent. AlphaDDA has the potential to be applied to any game where deep neural network can estimate values from state.

This paper is well-written and presented, and the authors address a novel and important application in board game. For example, in Go, even for professional players, playing directly against an AI can quickly lose its fun. Moreover, the experiment code is publicly available so that it can be better studied by the game AI community.

I would like to mention some comments and questions.

Experimental design

1. In the experiments, 40 games were played for each AI agents. It would be better if the number of games could be increased appropriately, as this makes the results more convincing.

2. line 154, what is “difference in the win and loss rates of AlphaDDA1 against the other AIs is minimal”. AlphaDDA2 and AlphaDDA3 have similar expressions. What would be the effect on the experimental results if this parameters were to change? In other words, are these parameters sensitive to the validity of the experimental results?

Validity of the findings

3. If some representative examples from games played by AlphaDDA against other AIs can be selected and shown (such as game record), it can further visualize the advantages and practicality brought by DDA.

---

## Round 0.2 · Minor Revisions

Review: AlphaDDA: game artificial intelligence with dynamic difficulty adjustment using AlphaZero

Summary:
The author proposes the method AlphaDDA as an extension to AlphaZero in order to reduce the playing strength of AlphaZero.
It comes as three variants AlphaDDA1, AlphaDDA2, and AlphaDDA3.
AlphaDDA1 changes the number of simulations of the MCTS
AlphaDDA2 uses dropout in order to add stochasty for move selection.
AlphaDD3 adopts the new UCT score depending on the value
The topic of the paper is relevant and interesting as having perfect playing AI system is unsatisfactory for human player to learn from.

Pros:
The overall structure seems fitting and concise.
Source code is also provided for readers who want to replicate or extend your work.
It also becomes clear that author has spent some work for both implementing the algorithms and the paper.

Cons:
The paper is general lacks a bit of language quality of a scientific paper and precision.
To elaborate this point, I will give a few examples from the abstract:
"In other words, the AI player becomes too strong as an opponent of human players."
Alternative formulation:
"In other words, the AI system surpasses the level of a strong human expert player."
"In order to entertain human players ..."
-> the word "entertain" seems a bit unsuitable here.
alternative: "In order for the AI system to become a suitable training partner"
"6x6 Othello, which is Othello using a 6x6 size board"
-> the fact that 6x6 Othello is played on a board of size 6x6 is obvious and not required to be stated here.
"with the other AI agents"
-> "with other AI agents"
(the word "the" is not needed here
"This study shows that AlphaDDA achieves to balance its skill with the other AI agents except for a random player."
-> This appears to be the main contribution of the paper.
In my opininion you could elaborate this a bit further using an additional sentence.
AlphaDDA's DDA ability is derived from the accurate estimation of the value
-> The word "derived" doesn't fully fit here. I'd choose "based".
"for any games in that the DNN can estimate the value from the state."
The condition on where you can apply this system appears too broad to me.
It lacks a bit precision. Is it applicable on all zero-sum games? Single- and two-player, multi-player games?
Is it applicable to discrete and continious action spaces?
What is also a bit confusing in the abstract is switching between "I" and "we".
I'd suggest to only use "we" instead
"AlphaDDA estimates the value of the game state from the board state using the
DNN. AlphaDDA adjusts its skill to the opponent by changing its skill according to the estimated value."
The two statements are a bit too ambigious.
What exactly is AlphaDDA doing?

Other:
The main contribution of the paper seems rather weak and self-explainatory.
"In order not to beat the opponent, the AlphaDDA is required to select a worse move."
This statement lacks detail.
Of course you could instead of playing the best move sample a different move.
But what kind of move instead and why?
I would suggest to change the title of the paper.
e.g. s.th. like
AlphaDDA: Strategies for Adjusting the Playing Strength of a Fully Trained AlphaZero System to a Suitable Human Training Partner.

Suggestion:
Revision of the paper, both in terms of the linguistic quality and the conciseness and precision of the contributions.

Reviewer 1 ·

Basic reporting

I am satisfied that authors provide proper explanations and the new version with corresponding changes according to my previous comments.
please adjust the size of tables 11, 12, 13 etc.
In addition, some recent work on AlphaZero should be cited.
Wang, H., Preuss, M., Plaat, A. (2020). Warm-Start AlphaZero Self-play Search Enhancements. In: , et al. Parallel Problem Solving from Nature – PPSN XVI. PPSN 2020. Lecture Notes in Computer Science(), vol 12270. Springer, Cham. https://doi.org/10.1007/978-3-030-58115-2_37
Wang, H., Preuss, M., Plaat, A. (2021). Adaptive Warm-Start MCTS in AlphaZero-Like Deep Reinforcement Learning. In: Pham, D.N., Theeramunkong, T., Governatori, G., Liu, F. (eds) PRICAI 2021: Trends in Artificial Intelligence. PRICAI 2021. Lecture Notes in Computer Science(), vol 13033. Springer, Cham. https://doi.org/10.1007/978-3-030-89370-5_5

Experimental design

good enough now

Validity of the findings

conclusions are well stated and evidenced by the experimental results

Reviewer 2 ·

Basic reporting

no comment

Experimental design

no comment

Validity of the findings

no comment

Additional comments

All my concerns have been addressed. I recommend this paper for publication.

---

## Round 0.3 · accepted · Accept

All reviewers' comments have been well addressed. The paper can be accepted.